# The International Classification of Functioning, Disability and Health to map leprosy-related disability in rural and remote areas in Indonesia

Luh Karunia Wahyuni [1][ʘ]*, Nelfidayani Nelfidayani[1ʘ], Melinda Harini[1ʘ], Fitri Anestherita[1ʘ], Rizky Kusuma Wardhani[1ʘ], Sri Linuwih Menaldi[2], Yunia Irawati[3], Tri Rahayu[3], Gitalisa Andayani[3], Hisar Daniel[3], Intan Savitri[1], Petrus Kanisius Yogi Hariyanto[1], Isabela Andhika Paramita[1,4]

1 Department of Physical Medicine and Rehabilitation, Faculty of Medicine Universitas Indonesia, dr. Cipto Mangunkusumo Hospital, Jakarta, Indonesia, 2 Department of Dermatology and Venereology, Faculty of Medicine Universitas Indonesia, dr. Cipto Mangunkusumo Hospital, Jakarta, Indonesia, 3 Department of Ophthalmology, Faculty of Medicine Universitas Indonesia, dr. Cipto Mangunkusumo Hospital, Jakarta, Indonesia, 4 Genomik Solidaritas Indonesia (GSI) Laboratory, Jakarta, Indonesia

ʘ These authors contributed equally to this work.
* luhkwahyuni@gmail.com

**Data Availability Statement:** All relevant data are within the manuscript and its supporting information files.

## Abstract

The International Classification of Function, Disability, and Health (ICF—WHO, 2001) recognizes several dimensions of disability, such as body structure and function (and impairment thereof), activity (and activity restrictions) and participation (and participation restriction) and their interactions with contextual factor (personal and environmental). In this study, we map and analyse the relationship between the components of ICF in leprosy patients from two rural areas in Indonesia: Lewoleba (East Nusa Tenggara) and Likupang (North Minahasa). This study was part of a community outreach program by the KATAMA-TAKU team from Universitas Indonesia. The body structure was graded using the WHO hand and feet disability grade and the number of enlarged nerves, while the body function was measured by the Jebsen Taylor Hand Function Test (JTT) and Timed-up and Go (TUG). Activity limitation and participation restriction were measured using the Screening Activity Limitation Safety Awareness (SALSA) Scale and Participation Scale (P-scale), respectively. There were 177 leprosy patients from the two regions and 150 patients with complete data were included in the analysis. We found 82% (95% CI: 75.08%-87.32%) of subjects with multibacillary leprosy, 10.67% (95% CI: 6.67%-16.62%) of subjects with grade 2 WHO hand disability, and 9.33% (95% CI: 5.64%-15.06%) of subjects with grade 2 WHO foot disability. Assessment using the SALSA Scale showed 29.33% of subjects with limitation activity and 11.33% with participation restriction. Age was shown to have positive correlations with SALSA, JTT, and TUG. Inter-dimensional analysis showed that the SALSA scale had significant positive correlations with the number of nerve enlargements, P-scale, JTT, and TUG. SALSA scores of grade 2 WHO hand and foot disability were also significantly higher than grades 1 and 0. The participation scale also had a positive correlation with JTT but not TUG. Hand disability seemed to affect societal participation while foot did

**Funding:** This study was funded by grant from Directorate of Research and Development, Universitas Indonesia under International-Indexed Publication Grant (PUTI) 2022 (Grant No: NKB-1402/UN2.RST/HKP.05.00/2022), with the recipient of our author LKW. Funder data could be accessed from https://ppm.ui.ac.id. The funders had no role in study design, data collection and analysis, decision to publish, or preparation of the manuscript.

**Competing interests:** The authors have declared that no competing interests exist.

not. We used the ICF to describe and analyse dimensions of leprosy-related disability in Indonesia.

## Author summary

Disability is the long-term outcome of untreated leprosy or Hansen's disease, which is caused by peripheral nerve invasion of the *Mycobacterium leprae*. It is a serious and life-limiting complication for leprosy patients. Currently, seven regions in Indonesia have not yet achieved the state of disease elimination. In addition, Indonesia has not succeeded in achieving the sub-target of the rate of new cases with grade 2 disability in 2023. The concept of disability and health including the body structure, body function, personal and environmental domains, activity limitation, and participation restriction, is provided by the ICF. Through the framework, we could assess the aspects of disability and its relationship to one another in leprosy patients. Furthermore, the early detection, screening, and management programs for leprosy have not included aspects of disability and rehabilitative measures for restoration. Thus, the purpose of this study is to find the impact of leprosy-related disability on the patients' body function, activity, and participation. We reported 150 leprosy patients from two rural areas of Indonesia and found significant relationships between specific indicators, which would be useful for rehabilitative management programs in the future.

## Introduction

Infiltration of *Mycobacterium leprae* into Schwann cells in leprosy can cause peripheral nerve inflammation and subsequent progressive loss of nerve fiber function or neuropathy [1]. This condition mainly affects the hands (44.45%), feet (39,76%), and face (15,74%) [2]. The subsequent loss of sensory; motor; and autonomic nerve function would appear as loss of thermal, nociceptive, and pressure senses; muscle paresis; and dryness of the skin, respectively. Occasionally, delayed emergence of skin lesions and sensory impairment may also occur, termed as silent neuropathy [1]. When left untreated, these conditions can develop into visible physical deformities (lagophthalmos, severe visual impairments, ulcers, clawing, and shortening of digits in the extremities), which the World Health Organization (WHO) classifies as grade 2 disabilities [3]. Unfortunately, the majority of newly diagnosed leprosy patients have already had deformities [2].

Disabilities in leprosy patients cause physical disability, psychological disturbances, and extensive loss of manpower and economic loss to society [3]. Assessment of disability in leprosy patients is a very important factor in the evaluation of the effectiveness of a leprosy elimination program [3]. Disability can be perceived as any impairment of a person's health and mind condition that causes the person to have functioning difficulty at the body, person, or societal levels, in any life domain [4,5]. The International Classification of Functioning, Disability, and Health (ICF) recognizes dimensions of disability, which include body structure and function (along with each impairment), activity and limitations, as well as participation and restrictions. Environmental aspects of the individual are also emphasized as a major predictor of disability outcomes thereby shifting the paradigm of disability as a mere 'medical' or 'biological' dysfunction [4].

Along with India and Brazil, Indonesia is one of the major leprosy contributors in the world. In the 2021 global leprosy update by WHO, the majority of new case detection came from Southeast Asia and more than 10,000 cases were found in Indonesia [6]. According to the Indonesian Ministry of Health report, leprosy cases are still increasing annually, from 5 cases per 100.000 population in 2021 to 5.5 cases per 100.000 population in 2022 [7]. Recent reports have shown that the number of leprosy patients with grade 2 WHO disability has widely ranged from 6.31% to 68.2% [2,8,9]. As of 2021, there were 12,230 registered leprosy cases in Indonesia, with a new case detection rate of 4.03 per 100,000 population and 10,976 new cases reported. The proportion of new leprosy cases with grade 2 deformities was determined to be 2.46 per 1,000,000 population, which still surpasses the WHO's target of 0.92 per 1,000,000 population by 2023 [10,11]. However, studies related to the burden of leprosy are fewer than those related to endemicity [12]. Not to mention that research about the specific challenges encountered by leprosy-related disability patients in Indonesia is very scarce. These data are urgently needed for initiating sustainable rehabilitative measures for disability prevention and for improving the patients' well-being.

The objectives of this study are to report leprosy cases from rural and remote areas in Indonesia, map the functional disabilities of individuals diagnosed with leprosy in Likupang and Lewoleba, Indonesia, using ICF, and to find the correlation between each domain of ICF in Indonesian population. We believe that there was an intercorrelation among aspects of ICF in the Indonesian leprosy population.

## Materials and methods

### Ethical aspects

The study protocol was approved by the Ethics Committee of the Faculty of Medicine, Universitas Indonesia under ethical clearance ND-454/UN2.F1/ETIK/PPM.00.02/2022 and was designed in consideration of the principles proposed by the Helsinki Declaration. Written formal consent was obtained directly from participants.

### Study design, period, and location

This cross-sectional data was obtained from a community outreach program by KATAMA-TAKU that was carried out in Lewoleba, Lembata, East Nusa Tenggara (July 2022) and Likupang, North Minahasa, North Sulawesi (August 2022). The KATAMATAKU team is a multidisciplinary (expert) team made up of dermatologists, ophthalmologists, and physical medicine and rehabilitation (PMR) specialists from the Faculty of Medicine Universitas Indonesia in Jakarta, Indonesia. The word KATAMATAKU is an Indonesian acronym for '*Identifikasi* *tanda-tanda* *mata*, *ekstremitas dan kulit pada kusta*' or identification of ocular, extremities, and dermatological signs in leprosy. The collaborative team often visits remote sites with higher leprosy-population density to do surveillance and charitable community engagements with leprosy patients and survivors. The outreach program consisted of physical examination, consultation, and treatment for leprosy patients.

The study participants from Likupang, North Minahasa, and Lewoleba, Lembata were those who attended the events held in July 2022 and August 2022, respectively. The individuals participating in our investigation received their diagnoses following the diagnostic criteria for leprosy established by the World Health Organization (WHO). Present-day leprosy diagnosis relies on the identification of at least one of the three primary diagnostic indicators: (i) pronounced sensory loss within a pale (hypopigmented) or reddish skin region; (ii) discernible thickening or enlargement of a peripheral nerve accompanied by sensory loss and/or muscle

weakness innervated by that nerve; or (iii) the detection of acid-fast bacilli in a slit-skin smear. [13].

Additionally, we also contacted the local public health officers a month prior to the event and the affiliated regional hospitals and primary health care facilities notified the residing population nearby and registered leprosy patients. Patients were referred by St. Damian Hospital (Lewoleba) and the public health office (Likupang) to attend our community outreach events in the respective months (July and August 2022).

Lewoleba, which is the capital of Lembata Island, East Nusa Tenggara, is in the Nubatukan district. Lewoleba has an area of 165.64 KM$^2$ whilst Lembata spans over 1.266,4 KM$^2$. Lembata Island is divided into nine districts and has an overall tropical climate with a long drought. Lembata has relatively high humidity at around 80% [14]. Nubatukan district has the highest population density at 50.984 or 37% of the total population of the island. Around 11.4% of the total population are aged 0–4 years. There are three hospitals and one public health center in Nubatukan [15]. The population in Lembata mostly works in agriculture, forestry, hunting, and fisheries sectors [15]. East Nusa Tenggara is considered as one of the provinces in Indonesia with high leprosy incidents, with around 7 per 100.000 people and a high chance of getting disability due to leprosy about 27.78% [15,16].

Likupang region is further divided into three subdistricts: East Likupang, West Likupang and South Likupang. All of them are coastal sub-districts located at 1.6720˚ N, 125.0553˚ E, 0 meters above the sea level, with a total area of 298.27 square kilometers (KM$^{2)}$ [17–19]. Since Likupang is located near the seashore, it has high humidity, reaching 80–90% on average [20]. In 2022, the region had 44.660 inhabitants, with average population growth rate of 1.39% between the years 2020 and 2022. South Likupang is the subdistrict with the highest population density, while East Likupang is the lowest [21]. According to data from the Ministry of Health in 2021, North Sulawesi is one of the Provinces in Indonesia with high leprosy cases, with detection rate of 11 per 100,000 inhabitants in 2018 [22,23]. As of 2022, North Sulawesi is one of the 7 provinces where leprosy has not yet been eliminated. Likupang itself has 4 public health centers, 3 with inpatient care, as well as 3 clinics spread over three subdistricts [17–19]. The map of Lewoleba, Lembata, and Likupang, North Minahasa, can be found in **Fig 1**.

## Study protocol and population

In both Likupang and Lewoleba, information was gathered through the examination of all leprosy patients. All subjects who agreed to participate in the study filled out the informed consent form. For patients below 18 years old, the informed consent forms were filled out by parents or legal guardians. The participants were then provided with a set of documents consisting of SALSA and P-scale questionnaires. Subsequently, patients were directed to independently complete these forms, with the option of seeking assistance from staff members for clarifications if needed. Additionally, they were encouraged to use only their initials, rather than their full names, to ensure the confidentiality of their identities. These completed forms were then collected and securely stored at the University of Indonesia. Our minimum requirement of samples was calculated using the expected correlation coefficient [24] as follows;

$$N = \frac{\left(Z_{\frac{\alpha}{2}} + Z_{1-\beta}\right)^2}{1/4\left[\log\left(\frac{1+r}{1-r}\right)\right]} + 4$$

with $Z_{\frac{\alpha}{2}} = 1.96$, $Z_{1-\beta} = 0,84$, and r = 0.8. We discovered that the required N = 33.925 or 34 patients.

The inclusion and exclusion criteria for the data used in our research are the following:

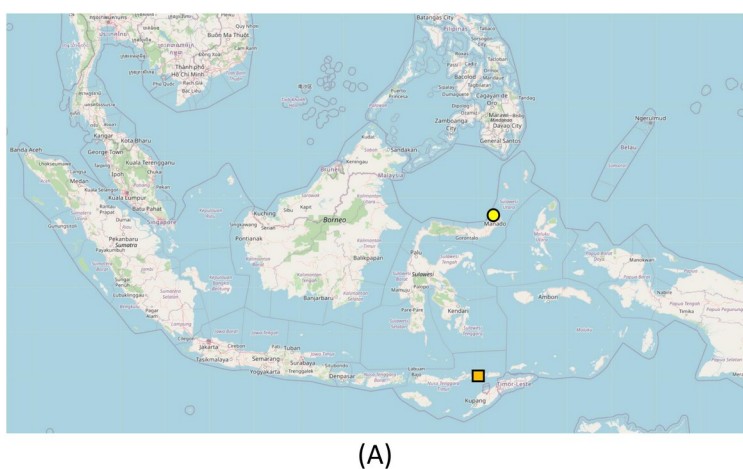

(A)

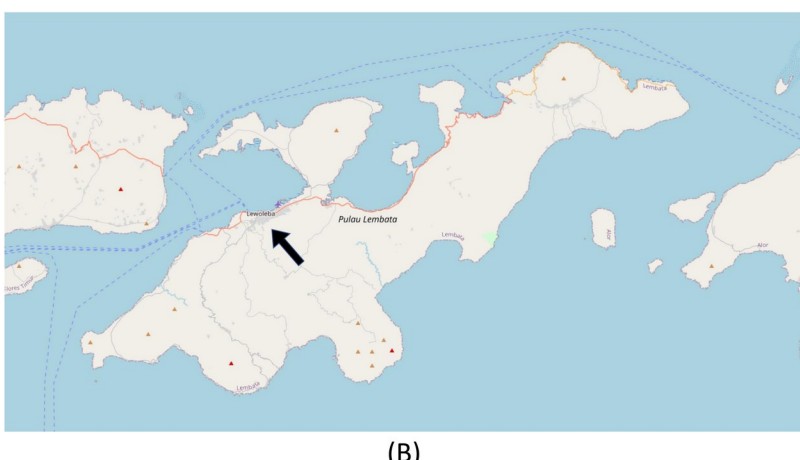

(B)

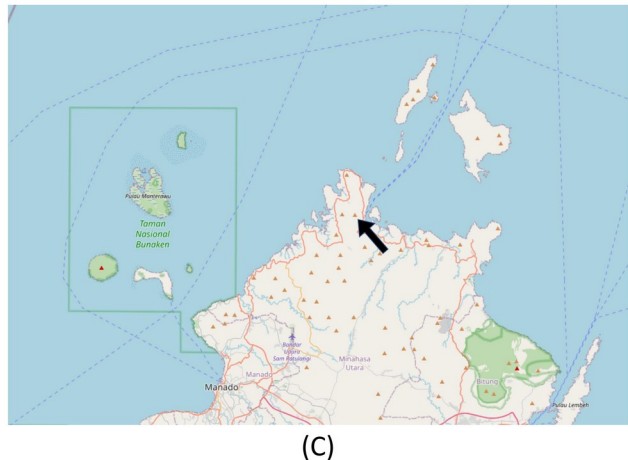

(C)

**Fig 1. Map of Lewoleba (Lembata, East Nusa Tenggara) and Likupang (North Minahasa, North Sulawesi), Indonesia.** (A) Map of Indonesia, scaled 1: 8000000 (2599x1527) (available from: https://www.openstreetmap.org/#map = 5/0.637/117.466&layers = G, Geo URI: geo:1.779,114.917?z = 5), yellow circle indicates Likupang, North Minahasa and orange rectangle indicates Lewoleba, Lembata. (B) Map of Lembata Island (East Nusa Tenggara), scaled 1: 273895 (1186 x 697) (available from: https://www.openstreetmap.org/#map = 10/-8.4078/123.6017&layers = G, Geo URI: geo:-8.3732,123.4369?z = 10), black arrow indicates Lewoleba. (C) Map of North Minahasa (North Sulawesi), scaled 1: 276738 (1174 x 690) (available from: https://www.openstreetmap.org/?mlat = 1.6961&mlon = 125.0011#map = 11/1.6961/125.0011&layers = G, Geo URI: geo:1.6960,125.0011?z = 13), black

arrow indicates Likupang. The map contains information from OpenStreetMap (OSM) and OpenStreetMap Foundation (accessed online on 16[th] of October 2023), Manual markings were done to locate Lewoleba and Likupang (Fig 1).

## Inclusion criteria

1. The patients are diagnosed with leprosy according to WHO criteria and referred from local hospitals and primary health care.

2. The study population consists of local residents.

3. The patient provided written consent to take part in this research.

## Exclusion criteria

1. Patients with neuropathic disabilities unrelated to leprosy.

2. The presence of pre-existing disabilities.

3. The presence of mental disabilities.

4. Patients who refused to undergo examinations for body function (Jebsen-Taylor Hand Function Test (JTT) and timed up and go (TUG) test), activity limitation (SALSA), and participation restriction (P-scale).

Age, gender, and background were among the basic demographic data collected. Each participant filled out the SALSA questionnaire as well as the participation scale (P-scale) and went through the entire assessment process, including the timed up-and-go (TUG) test and the Jebsen-Taylor Hand Function Test (JTT). The SALSA questionnaire measures the activity limitation of the patients and consists of 20 questions related to the ability to do daily activities and the three domains used in this study are mobility, self-care, and work [25]. The participation scale consists of 18 questions related to the social participation of the patients (e.g. working, socializing with other people) [26]. These questionnaires were available in Indonesian language [25,26] and were performed in Indonesian population prior to the study [8,27,28].

The study variables or assessment tools along with each description are listed in **Table 1**. The outcomes were analysed by the International Classification of Functioning, Disability, and Health (ICF) domains as listed in **Fig 2**.

**Analysis of results and statistics.** Microsoft Excel was used to store, organize, clean raw data and perform F tests for equal variance. Statistical analysis was performed using IBM Statistical Package for Social Sciences (SPSS) Statistics Software version 25.0 (IBM Corp., Armonk, NY) and figures were generated using GraphPad Prism version 10.0.2 (GraphPad Software, San Diego, CA). Demographics of patients and the 95% confidence intervals were calculated with RStudio using 'prop.test' function for test of one proportion. Normality and F tests were performed to all sets of data. For the comparison test between two groups, non-normal groups were analysed non-parametrically using Mann-Whitney U test and presented with median $\pm$ range. Normally distributed groups with equal and non-equal variances were compared using independent $t$ test (presented with mean $\pm$ standard deviation (SD)) and Welch's $t$ test (presented with mean), respectively. For the comparison between more than two groups, groups with non-normal distribution were analysed using Kruskal-Wallis test. Bivariate analyses using Pearson's and Spearman's correlation tests were performed in normally and non-normally distributed variables, respectively. Correlation coefficients between -0.3 to 0.3 were

**Table 1. Assessment Tools according to the domains of International Classification of Functioning, Disability, and Health (ICF).**

| Item | Content | Evaluation | Note |
|---|---|---|---|
| *Personal factors* | | | |
| • Age | | Identity card | <18 years old, 18–59 years old, ≥60 years old |
| • Gender | | Identity card | Female, male |
| • Formal education level | | Question to the patient | • Uneducated<br>• Primary Education<br>• Middle School<br>• Secondary Education<br>• Higher Education |
| *Health condition* | | | |
| • Diagnosis of   Leprosy | | Anamnesis related to history of the disease, type of leprosy, and medication | Paucibacillary, multibacillary |
| • Number of person(s) living together with the same diagnosis | | Anamnesis of any contacts within family members and whether they live together or separately with the patient | Number of family member with similar symptoms or diagnosis |
| *Body structure* | | | |
| - Physical Examinations | Abnormalities and deformities in the skin and extremities | - Inspection of lesions and deformities<br>- · Motor and sensory nerve function | Number of nerve thickening. |
| - The WHO classification of physical disability in leprosy [30]. | | The presence of nerve impairments or visible deformity | Grade 0: no disability (no loss of sensitivity and no visible deformity or damage to the eyes, hands or feet)<br>Grade 1: only disability (a loss of sensitivity without visible deformity or damage to the eyes, hands or feet)<br>Grade 2: the presence of visible deformity or damage to the eyes (lagophthalmos, iridocyclitis, corneal opacities, severe visual impairment), hands (claw hands, ulcers, absorption of the digits, thumb-web contracture and swollen hand), feet (plantar ulcers, footdrop, inversion of the foot, clawing of the toes, absorption of the toes, collapsed foot and callosities) |
| *Body function* | | | |
| • The Jebsen-Taylor Hand Function Test (JTT) [31]. | A standardized and objective measure of fine and gross motor hand function using simulated activities of daily living (ADL) | A series of seven subtests representing fine motor, non-weighted and weighted hand function in ADL, which includes: writing, simulated page-turning, lifting small objects, simulated feeding, stacking, and lifting large, lightweight, and heavy objects. | The subtests are scored by recording the number of seconds required to complete each test<br>Total score is the sum of time taken for each sub-test, which are rounded to the nearest second. Shorter times indicate better performance. |
| • Timed Up and Go Test (TUG) [32]. | To determine the patient's fall risk and measure the progress of balance, sit to stand and walking | Assessed domains<br>• Balance<br>• Gait<br>• Walking speed | Patients performed the test one time, and if a clear error was made, they were asked to repeat the TUG. |
| *Activity* | | | |
| • The Screening Activity Limitation Safety Awareness (SALSA) Scale [25,33]. | Includes 20 daily activities questions related to the three areas of mobility, self-care, and work. | The ability of patients to do basic daily activities such as walking, self-care, carrying heavy objects and manipulating small objects. | SALSA's Indonesian edition was used for this investigation. The total score ranges from 0–80, with the result classified as:<br>• no substantial limitation (0–24)<br>• mild limitation (25–39)<br>• moderate limitation (40–49)<br>• severe limitation (50–59)<br>• extreme limitation (60–80) |
| *Participation* | | | |

*(Continued)*

**Table 1.** (Continued)

| Item | Content | Evaluation | Note |
|------|---------|-----------|------|
| • Participation Scale (P-Scale) [34]. | Consists of 18 items to measure (social) participation for use in rehabilitation, stigma reduction and social integration programmes. | Assesses domains of participation including:<br>• Communication<br>• Mobility<br>• Self-Care<br>• Domestic Life<br>• Interpersonal Interactions<br>• Major life areas (work, education, etc)<br>• Community, Social and Civic | The P-scale total score varies between 0–90, with the result classified as:<br>• No significant restriction (0–12)<br>• Mild restriction (13–22)<br>• Moderate restriction (23–32)<br>• Severe restriction (33–52)<br>• Extreme restriction (53–90) |

considered independent [35]. Heatmap was created using RStudio using the 'lattice' package along with 'cor' and 'levelplot' function. Multivariable regression analyses using SPSS were used to adjust for confounders in statistically correlated results to prevent overestimation. The confounders considered in the multivariable regression analyses were variables with quantitative values such as age (year), length of disease (month), length of complaint before diagnosis (month), SALSA scales (including -mobility, -work, and -dexterity), P-scale, JTT in dominant and nondominant hands (seconds), and TUG (seconds). Categorical variables were analysed using Fisher's Exact and Chi-Square tests, and multivariate analyses with binary logistic regression was further performed to statistically significant results with subsequent adjustment of confounders (presented with Odds Ratio (OR) and p-values). Confounders considered in this

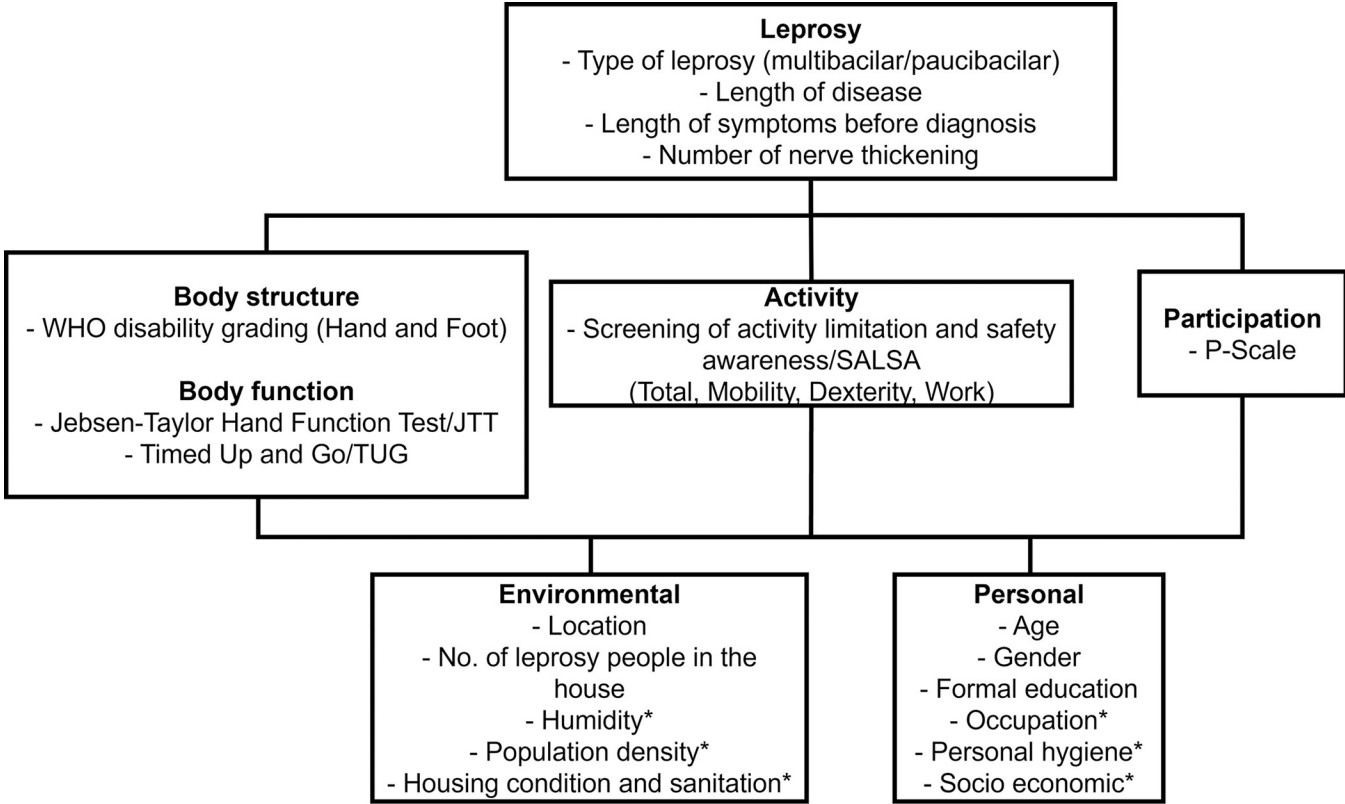

**Fig 2. Application of the International Classification of Functioning, Disability and Health (ICF) domains in the study.**[29] The domains of ICF have been adjusted to the availability of data within both populations in Likupang and Lembata. SALSA, screening of activity limitation and safety awareness; JTT, Jebsen-Taylor hand function test; TUG, timed up and go test.

study include age (juvenile or $\leq$ 18 years, adult (18–59 years), and old ($\geq$ 60 years)), gender (male and female), educational background (uneducated, primary education, middle school, secondary education, and higher education), location (Likupang and Lewoleba), occurrence of nerve thickening (none and present), length of disease ($<$ 24 (median) months and $\geq$ 24 months), length of complaint before diagnosis ($<$ 6 (median) months and $\geq$ 6 months), type of leprosy (paucibacillary and multibacillary), hand and foot WHO disability (grade 0, grade 1, and grade 2), SALSA categories (no limitation, mild limitation, moderate limitation, severe limitation, and extreme limitation), P-scale categories (no restriction, mild restriction, moderate restriction, severe restriction, and extreme restriction), JTT in dominant hand ($<$20.94 (median) s and $\geq$ 20.94 s) and nondominant hand ($<$24.88 (median) s and $\geq$ 24.88 s), and TUG ($<$ 13.5 s and $\geq$ 13.5 s). Probability of $<$0.05 is considered statistically significant.

## Results

The total number of patients examined during the events were 177 patients with leprosy. After excluding patients with incomplete data, 150 patients (107 in Lewoleba and 43 in Likupang) were included in the analysis. Mean $\pm$ standard deviation (SD) of the patients' age (years) was 44.71 $\pm$ 18.7, whilst median(s) $\pm$ range of the patients' length of disease (months) and length of symptoms before diagnosis (months) are 24 $\pm$ 720 and 6 $\pm$ 228, respectively. Majority of the patients were between 18–59 years old (64.67%), male (69.33%), graduated from elementary/ primary schools (53.33%), did not have hand (73.33%) or foot (73.33%) disability, have no limitation nor restriction in functional activity (70.67%) and participation (88.67%), respectively, had been diagnosed with multibacillary type (82%), had none other family members living together diagnosed with leprosy (68.67%), have had leprosy for longer than 24 months (54.67%), and had been experiencing symptoms for more than six months before being diagnosed (48.67%) (**Table 2**).

Analysis of the data was done according to ICF, introduced by the WHO in 2001. We divided the examinations according to several aspects. Age, gender, and types of leprosy are included into the patient's condition. The number of thickened nerves and WHO hand and foot disability grading represent body structure. Jebsen-Taylor Hand Function Test and TUG are parts of body function. The SALSA and Participation Scales represent activity and participation, respectively. Initially, we would like to find whether there are any differences in the patient's condition, body structure, body function, activity, and participation according to gender, age, education, and types of leprosy. All data were not normally distributed, and many sets of groups did not have equal variances. We found a significant difference in the number of thickened nerves between male and female (Mann-Whitney U test, median 0.00 vs 0.00, 95% CI of median difference: 0.00 to 0.00, p = 0.0021) as well as the number of nerve thickening occurrence in male compared to female (Fisher's exact test, p = 0.0041). Binary logistic regression showed OR of 3.626 (95% CI: 1.481 to 8.878, p = 0.005) in males compared to female for the development of nerve thickening. However, adjusted OR showed 3.048 (95% CI: 0.810 to 11.469, p = 0.099). There was also a significant difference in the frequency of multibacillary type infection in male and female (Fisher's exact test, p = 0.0384) with OR of 2.532 (95% CI: 1.078 to 5.948, p = 0.033) in males towards multibacillary type compared to females but adjusted OR of 2.018 (95% CI: 0.686 to 5.942, p = 0.202). There were significant positive correlations between age and SALSA scales including the mobility (r = 0.3965, 95% CI: 0.2478 to 0.5269, p$<$0.0001) and dexterity (r = 0.4738, 95% CI: 0.3351 to 0.5924, p$<$0.0001) aspects, JTT in both hands, and TUG as seen in **Table 3**. The R Square of the multivariable regression model with age as dependent variable is 33.2% (p$<$0.0001) for predictors as follows: length of disease, length of complaint before diagnosis, SALSA scales (Total, -Mobility, -Work,

**Table 2. Demographics of Leprosy Patients in Lewoleba (Lembata) and Likupang (North Minahasa), Indonesia (n = 150).**

| Characteristics | n | Proportions with 95% Confidence Interval (%) |
|---|---|---|
| *Age* | | |
| • ≤ 18 | 13 | 8.67 (5.135–14.26) |
| • 18–59 | 97 | 64.67 (56.74–71.86) |
| • ≥ 60 | 40 | 26.67 (20.24–34.26) |
| *Gender* | | |
| • Female | 46 | 30.67 (23.85–38.45) |
| • Male | 104 | 69.33 (61.55–76.15) |
| *Educational Background* | | |
| • Uneducated | 9 | 6 (3.19–11.01) |
| • Primary Education | 80 | 53.33 (45.37–61.13) |
| • Middle School | 21 | 14 (9.34–20.46) |
| • Secondary Education | 33 | 22 (16.12–29.28) |
| • Higer Education | 7 | 4.67 (2.28–9.32) |
| *WHO Hand Disability* | | |
| • Grade 0 | 114 | 76 (68.57–82.13) |
| • Grade 1 | 20 | 13.33 (8.80–19.70) |
| • Grade 2 | 16 | 10.67 (6.67–16.62) |
| *WHO Foot Disability* | | |
| • Grade 0 | 110 | 73.33 (65.74–79.76) |
| • Grade 1 | 26 | 17.33 (12.11–24.19) |
| • Grade 2 | 14 | 9.33 (5.64–15.06) |
| *SALSA Score* | | |
| • No limitation | 106 | 70.67 (62.94–77.36) |
| • Mild limitation | 41 | 27.33 (20.83–34.96) |
| • Moderate limitation | 2 | 1.33 (0.37–4.73) |
| • Severe limitation | 0 | 0 (0.00–2.50) |
| • Extreme limitation | 1 | 0.67 (0.12–3.68) |
| Participation Scale | | |
| • No restriction | 133 | 88.67 (82.60–92.80) |
| • Mild restriction | 10 | 6.67 (3.66–11.84) |
| • Moderate restriction | 6 | 4 (1.85–8.45) |
| • *Severe restriction* | 0 | 0 (0.00–2.50) |
| • Extreme restriction | 1 | 0.67 (0.12–3.68) |
| Jebsen-Taylor Hand Function Test (JTT) | | |
| Dominant Hand | | |
| • < 20.94 s | 75 | 50 (42.10–57.90) |
| • ≥ 20.94 s | 75 | 50 (42.10–57.90) |
| Non-dominant Hand | | |
| • < 24.88 s | 75 | 50 (42.10–57.90) |
| • ≥ 24.88 s | 75 | 50 (42.10–57.90) |
| Timed-up and go test | | |
| • < 13.5 s | 146 | 97.33 (93.34–98.96) |
| • ≥ 13.5 s | 4 | 2.67 (1.04–6.66) |
| Location | | |
| • Lembata | 107 | 71.33 (63.64–77.97) |
| • Likupang | 43 | 28.67 (22.03–36.36) |

*(Continued)*

**Table 2.** (Continued)

| Characteristics | n | Proportions with 95% Confidence Interval (%) |
|---|---|---|
| Number of Family Members with Leprosy Living Together | | |
| • None | 103 | 68.67 (60.86–75.55) |
| • 1 | 38 | 25.33 (19.05–32.85) |
| • 2 | 5 | 3.33 (1.43–7.57) |
| • >2 | 3 | 2 (0.68–5.71) |
| • Undisclosed | 1 | 0.67 (0.12–3.68) |
| Types of Leprosy | | |
| • Paucibacillary | 27 | 18 (12.68–24.92) |
| • Multibacillary | 123 | 82 (75.08–87.32) |
| Length of disease | | |
| • < 24 months | 63 | 42 (34.40–50.00) |
| • 24 months ++ | 82 | 54.67 (46.68–62.42) |
| • Undisclosed | 5 | 3.33 (1.43–7.57) |
| Length of complaint before diagnosis | | |
| • < 6 months | 66 | 44 (36.30–52.00) |
| • 6 months ++ | 73 | 48.67 (40.80–56.60) |
| • Undisclosed | 11 | 7.33 (4.14–12.65) |

-Dexterity), JTT in dominant and nondominant hands, and TUG. Age is a significant predictor of P-scale (B = -0.110 (95% CI: (-0.181)-(-0.039)), Beta = -0.291, p = 0.003) and JTT in dominant hand (B = 8.81 (95% CI: 1.45–16.18), Beta = 0.18, p = 0.019). Finally, we also found significant differences between levels of education in total SALSA scores, SALSA-Work scores, SALSA-Dexterity scores, JTT in both dominant and non-dominant hands, and TUG (**Fig 3A-3E**).

Next, we wanted to evaluate the environmental factors of ICF in our populations, so we divided the patients based on the areas of the community outreach; Likupang, North Sulawesi, and Lewoleba, West Nusa Tenggara. We found that there was a significant difference of mutibacillar leprosy occurrence between the population in Lewoleba and Likupang (Fisher's exact test, p = 0.0017). Binary logistics regression analyses showed OR of 4.241 (95% CI: 1.780 to 10.105, p = 0.001) for multibacillary occurrence in Lewoleba compared to Likupang. However, adjusted OR showed non-significant value of 1.843 (95% CI: 0.572 to 5.935, p = 0.305). The occurrence of nerve enlargement between the participants living in Likupang and Lewoleba were found to seemingly different using Fisher's exact test (p<0.0001) but not with binary logistics (p = 0.997). There were no significant differences of length of disease, length of

**Table 3. Significant bivariate analyses of ICF variables with quantitative values with age and their adjusted coefficients as predictors of age after multivariable linear regression analyses.**

| Other variables | Total SALSA | SALSA-Mobility | SALSA-Dexterity | JTT (dominant hand) | JTT (non-dominant hand) | TUG |
|---|---|---|---|---|---|---|
| Spearman's rho | 0.50 | 0.40 | 0.47 | 0.49 | 0.42 | 0.49 |
| 95% CI | 0.37–0.62 | 0.25–0.53 | 0.34–0.59 | 0.36–0.61 | 0.27–0.55 | 0.35–0.61 |
| p-value (bivariate analyses) | <0.0001 | <0.0001 | <0.0001 | <0.0001 | <0.0001 | <0.0001 |
| Unstandardized coefficients (B) | 1.10 | 1.20 | -0.19 | 0.01 | -0.00 | 0.01 |
| 95% CI | (-1.78)-3.98 | (-1.97)-4.37 | (-4.21)-3.84 | 0.00–0.01 | 0.00–0.00 | 0.00–0.01 |
| Standardized coefficients (Beta) | 0.30 | 0.14 | -0.02 | 0.23 | -0.05 | 0.11 |
| p-value (multivariable regression analyses) | 0.451 | 0.454 | 0.927 | 0.019 | 0.589 | 0.197 |

SALSA, screening of activity limitation and safety awareness; JTT, Jebsen-Taylor hand function test; TUG, timed up and go test; CI, confidence interval.

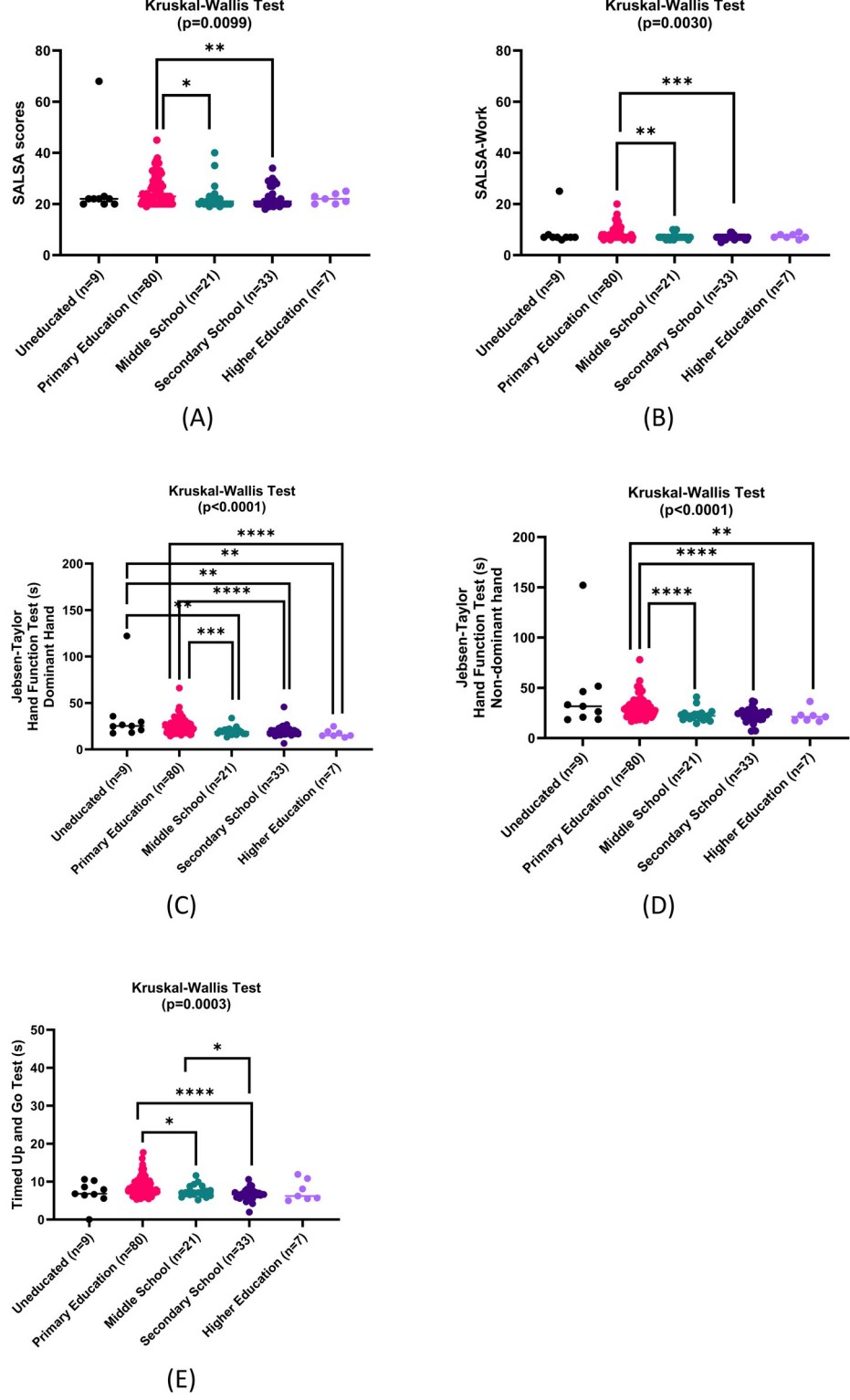

**Fig 3. Comparison of SALSA and P-scales according to different educational levels (n = 150).** (A) Total SALSA scores. Mann-Whitney U tests showed statistically different scores between those who attended primary education vs middle school (median $\pm$ range = 23 $\pm$ 26 vs 21 $\pm$ 21, 95% CI of median difference: -4.00 to 0.00, respectively, p = 0.0127), as well as those who attended primary education and secondary school (median $\pm$ range = 23 $\pm$ 26 vs 21 $\pm$ 16, respectively, 95% CI of median difference: -4.0 to -1.0, p = 0.0025). (B) SALSA-Work scores. Mann-Whitney U

tests yielded significant different scores between primary education vs middle school (median $\pm$ range = 7 $\pm$ 14 vs 7 $\pm$ 4, respectively, 95% CI of median difference: -1.0 to 0.0, p = 0.0092), as well as those who attended primary education and secondary school (median $\pm$ range = 7 $\pm$ 14 vs 7 $\pm$ 4, respectively, 95% CI of median difference: -1.0 to 0.0, p = 0.0003). (C) Jebsen-Taylor Hand Function Test in dominant hand. Significant differences were observed between uneducated (25.28 $\pm$ 104.6 s) vs middle school (19.06 $\pm$ 20.48 s, 95% CI of median difference: -11.33 to -1.29, p = 0.0091), secondary school (19.19 $\pm$ 39.19 s, 95% CI of median difference: -11.20 to -1.78, p = 0.0068), and higher education (14.97 $\pm$ 12.02 s, 95% CI of median difference: -16.72 to -2.67, p = 0.0033), as well as between primary education (23.71 $\pm$ 51.76 s) vs middle school (95% CI of median difference: -7.30 to -2.19, p = 0.0002), secondary school (95% CI of median difference: -7.00 to -2.56, p<0.0001), and higher education (95% CI of median difference: -11.91 to -3.19, p = 0.0004). (D) Jebsen-Taylor Hand Function Test in non-dominant hand. Mann Whitney U tests showed statistically significant differences between the group who only graduated from primary education (28.17 $\pm$ 61.40 s) and those who finished middle school (22.06 $\pm$ 26.31 s, 95% CI of median difference: -8.98 to -3.00, p<0.0001), secondary school (23.51 $\pm$ 29.88 s, 95% CI of median difference: -8.43 to -2.66, p<0.0001), and higher education (21.26 $\pm$ 19.82 s, 95% CI of median difference: -12.06 to -1.76, p = 0.0099). (E) Timed up and go test. Mann Whitney U tests showed significant differences between those who only attended primary education (8.015 $\pm$ 12.37 s) with those who attended middle school (7.110 $\pm$ 6.450 s, 95% CI of median difference: -1.69 to -0.04, p = 0.0348) and secondary school (6.650 $\pm$ 8.620 s, 95% CI of median difference: -2.34 to -0.80, p<0.0001). Unpaired $t$ test showed significant differences between those who attended middle school and secondary school (mean $\pm$ SD = 7.453 $\pm$ 1.516 s vs 6.585 $\pm$ 1.519 s, 95% CI: -2.606 to 3.847, p = 0.0454).

symptoms before diagnosis, SALSA scores including mobility, work, and dexterity, P-scale, JTT in both hands, and TUG between Likupang and Lewoleba. Most of the patients did not have any people in the household who were also diagnosed with leprosy at the time of the examinations (**Table 2**). Chi-square tests between the number of household members with leprosy and the other variables showed no significant differences.

To assess different domains in relation to body structure impairment, Kruskal-Wallis tests were performed on SALSA scores, P-scales, JTT, and TUG according to WHO hand and foot disability grades. We found significant differences of the medians of SALSA scale, P-scale, and JTT in patients with grade 0, 1, and 2 WHO hand disability, whilst only comparison of SALSA scales in patients with grade 0, 1, and 2 WHO foot disability yields significant results as seen in **Figs 4** and **5**.

Next, we wanted to see if the type of leprosy affected the other variables from the other domains of ICF. Comparison tests of body structure impairment, body function impairment, activity limitation, and participation restriction between types of leprosy; paucibacillary or multibacillary, were performed. We found no significant differences. However, Fisher's exact tests showed that the occurrence of nerve enlargements were significantly higher in multibacillary cases (p = 0.0024). Binary logistic regression analyses showed OR of 7.468 (95% CI: 1.69 to 32.997, p = 0.008) and adjusted OR of 29.782 (95% CI: 1.922 to 461.412, p = 0.015) for occurrence of nerve enlargement in multibacillary type compared to paucibacillary leprosy. Bivariate analysis showed that the length of disease, length of symptoms before diagnosis, and number of nerve thickenings were found to be independent to all other variables.

Finally, we wanted to see whether the variables in the body function domain were inter-correlated with the activity limitation and participation restriction domains in the study population (**Fig 6**). There were positive correlations within the total SALSA-scores and SALSA-Mobility, -Work, and -Dexterity. There was a positive and significant correlation between P-scale and SALSA-mobility scores. There were positive and significant correlations between total SALSA scores, SALSA-work, and -dexterity and the Jebsen-Taylor hand function test. We also found significant correlations between TUG and SALSA along with SALSA-mobility scores. There were positive and significant correlations between JTT in dominant and non-dominant hands, TUG and SALSA-Work, and TUG and JTT in both hands.

We further analyse whether any predictive relationships exist between the significantly correlated variables (body function, activity, and participation) using multivariable

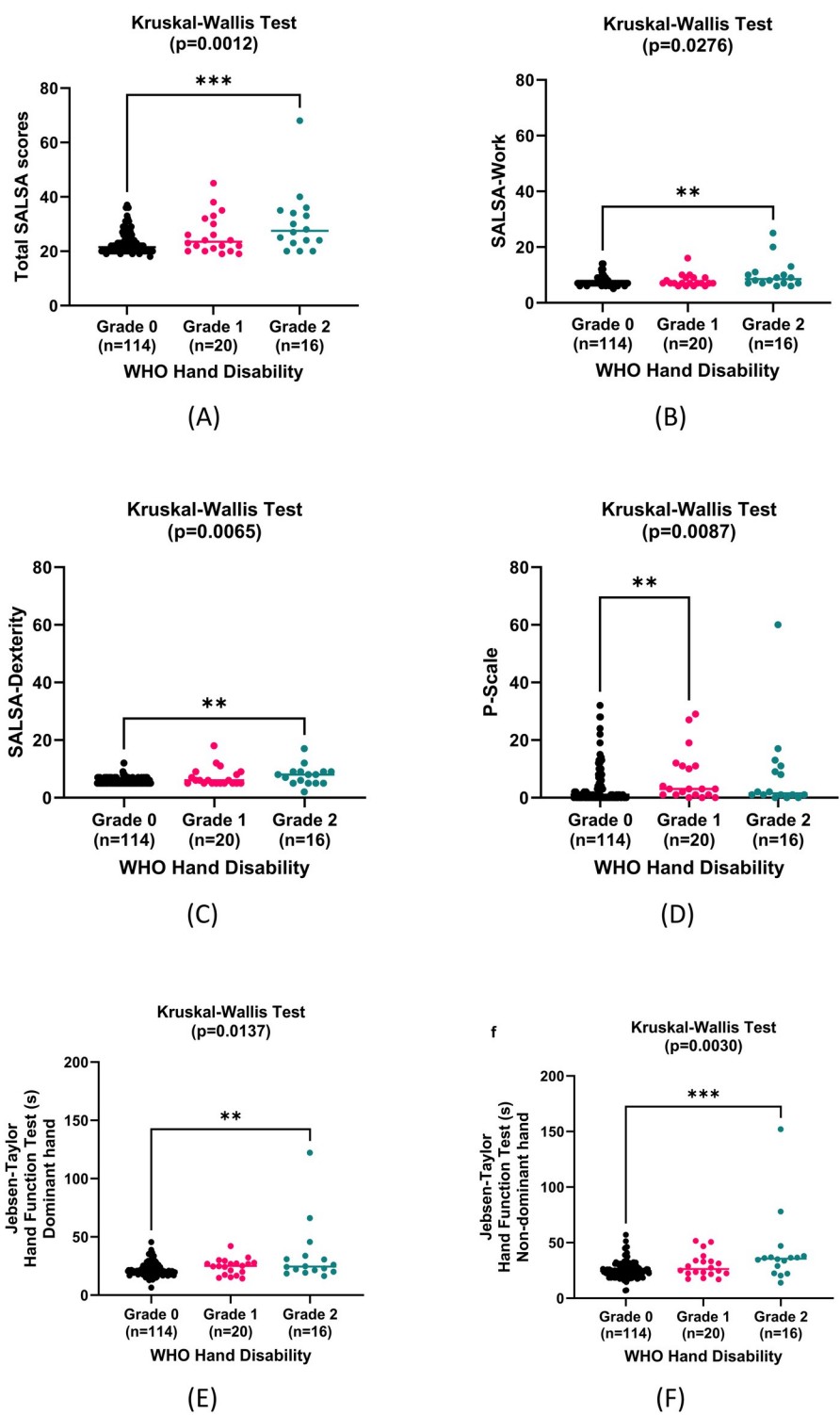

**Fig 4. Comparison of hand structure impairment according to hand function impairment, activity limitation, and participation restriction (n = 150).** (A) Total SALSA scores. Mann-Whitney U test showed significant differences between grade 0 and grade 2 (median $\pm$ range 21.50 $\pm$ 19 vs 27.50 $\pm$ 48, 95% CI of difference: 2.00 to 9.00, p = 0.0004). (B) SALSA-Work. Mann-Whitney U test showed significant differences between grade 0 and grade 2 (7 $\pm$ 9 vs 8.5 $\pm$ 19, 95% CI of difference: 0.00 to 2.00, p = 0.0058). (C) SALSA-Dexterity. Mann-Whitney U test showed significant differences between grade 0 and grade 2 (5 $\pm$ 7 vs 8 $\pm$ 15, 95% CI of difference: 0.00 to 3.00, p = 0.0018). (D) P-Scale. Mann-Whitney U test showed significant differences between grade 0 and grade 1 (1 $\pm$ 32 vs 3 $\pm$ 29, 95% CI of median

difference: 0.00 to 3.00, p = 0.0070). (E) JTT dominant hand. Mann-Whitney U test showed significant differences between grade 0 and grade 2 (20.5 ± 39.06 s vs 24.51 ± 105.8 s, 95% CI of median difference: 1.09 to 8.54, p = 0.0077). (F) JTT non-dominant hand. Mann-Whitney U test showed significant differences between grade 0 and grade 2 (24.37 ± 50.03 s vs 35.62 ± 138.2 s, 95% CI of median difference: 4.40 to 14.04, p = 0.0009).

regression analyses. We found that the JTT in both hands were not predictors of total SALSA scales, SALSA-work, and SALSA-dexterity, TUG not being a predictor for total SALSA scale and SALSA-mobility, as well as P-scale not being a predictor for SALSA-mobility. In turn, the SALSA scales were not predictive of JTT in both hands, TUG, and P-scale (**S17 Data**).

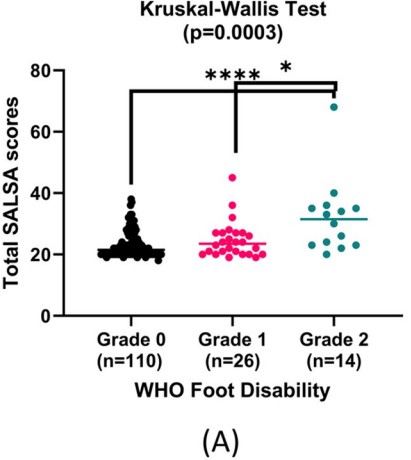
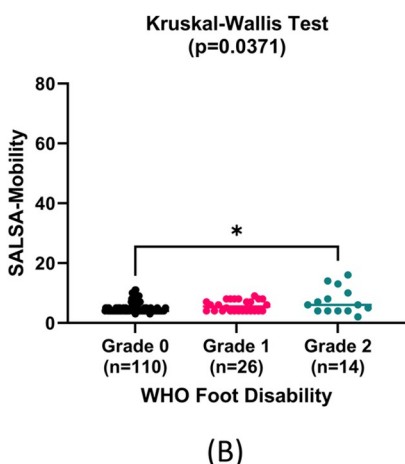

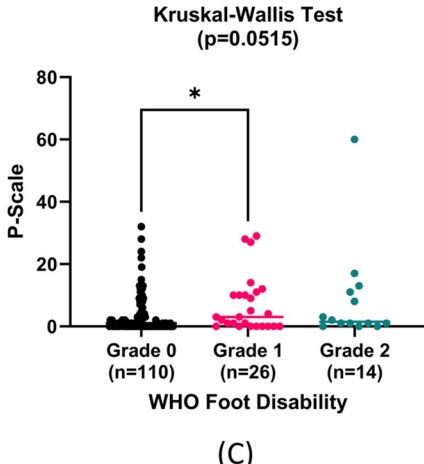

**Fig 5. Comparisons of foot structure impairment according activity limitation and participation restriction (n = 150).** (A) Total SALSA scores. Mann-Whitney U test showed significant differences between grade 0 and grade 2 (median ± range 21.25 ± 20 vs 31.50 ± 48, 95% CI of median difference: 3.00 to 12.00, p<0.0001) and between grade 1 and grade 2 (23.5 ± 26 vs 31.50 ± 48, 95% CI of median difference: 1.00 to 2.00, p = 0.0146). (B) SALSA-Mobility. Mann-Whitney U test showed significant differences between grade 0 and grade 2 (4 ± 8 vs 6 ± 14, 95% CI of median difference: 0.00 to 3.00, p = 0.0339). (C) P-scale. Mann-Whitney U test showed significant differences between grade 0 and grade 2 (1 ± 32 vs 3 ± 29, 95% CI of median difference: 0.00 to 3.00, p = 0.0413).

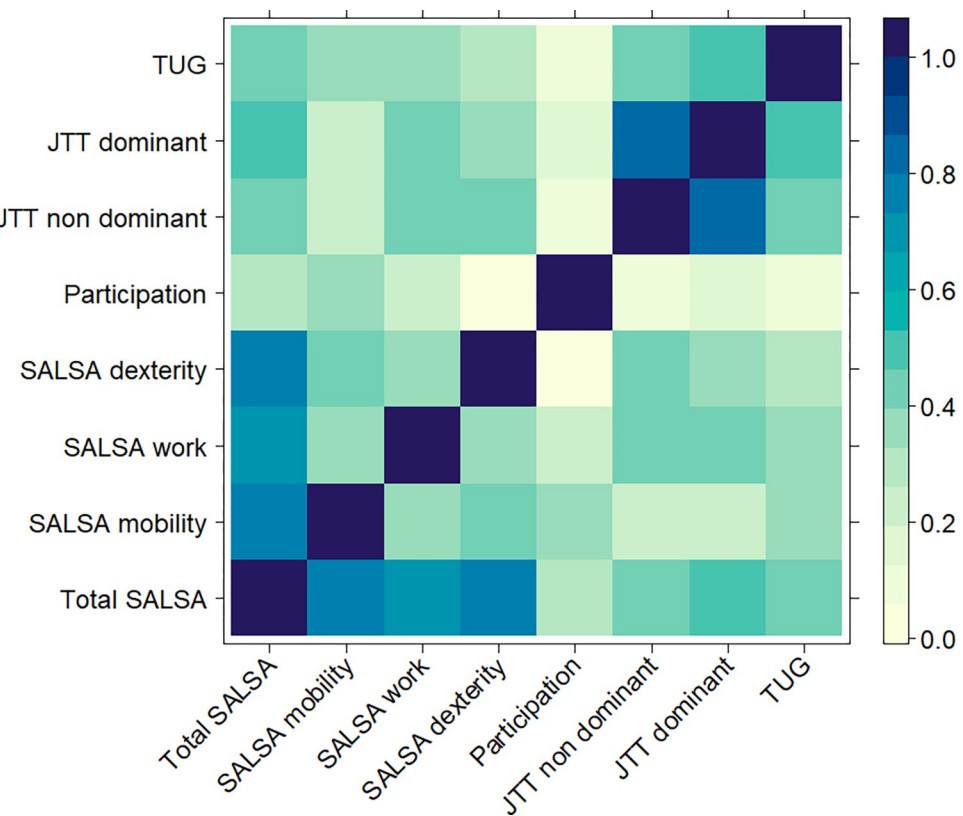

**Fig 6. Bivariate analyses of the variables in the body function, activity limitation, and participation restriction domains according to the International Classification of Functioning, Disability and Health (ICF) (n = 150).** Strongest correlations were observed between body function impairment (JTT) in dominant hand and total SALSA scores (Spearman's rho = 0.4659, 95% CI: 0.3261–0.5858, p<0.0001) and in non-dominant hand (Spearman's rho = 0.4296, 95% CI: 0.2849–0.5552, p<0.0001), followed by correlations between JTT and SALSA-work in dominant (Spearman's rho = 0.4236, 95% CI: 0.2781–0.5501, p<0.0001) and non-dominant (Spearman's rho = 0.3967, 95% CI: 0.2480–0.5271, p<0.0001) hands. There were also significant correlations between JTT in dominant and non-dominant hands with SALSA-dexterity (Spearman's rho = 0.3737, 95% CI: 0.2225–0.5074, p<0.0001 and Spearman's rho = 0.4212, 95% CI: 0.2755–0.5481, p<0.0001, respectively), between body function impairment (TUG) and activity limitation in relation to lower extremities (total SALSA (Spearman's rho = 0.4232, 95% CI: 0.2776–0.5497, p<0.0001) and SALSA-mobility (Spearman's rho = 0.3296, 95% CI: 0.1741–0.4690, p<0.0001)), and between participation restriction (P-scale) and SALSA-mobility (Spearman's rho = 0.3347, 95% CI: 0.1797–0.4735, p<0.0001)). SALSA, screening of activity limitation and safety awareness; JTT, Jebsen-Taylor hand function test; TUG, timed up and go test.

## Discussion

One of the purposes of this study is to utilize ICF to assess the functional disabilities experienced by individuals diagnosed with leprosy in Likupang, North Minahasa and Lewoleba, Lembata, Indonesia. ICF recognizes several dimensions of disability, including impairment in body structure and function, restrictions in activity, and limitations in participation. Additionally, the classification acknowledges the significant influence of physical and social environmental factors on the outcomes of disability. Even though ICF is very important in assessing a patient's level of disability, it is still not widely used in Indonesia, including for leprosy patients.

In this study, we found that hand and foot disability affected the activity limitation and participation restriction domains in this population. Hand disability also affects body function domains. Both participation restriction and activity limitation had a significant correlation to

one another. However, activity limitations also had a significant correlation with upper and lower body functions. Activity limitation and body functions were affected by age and educational levels. We also found that the locations of the population and the types of leprosy did not significantly affect disability progression. All in all, our results imply that the body structure and body function domain in ICF affected activity limitation and participation restriction and vice versa in leprosy patients [4].

Several personal factors for leprosy and leprosy-related disabilities have been studied. A meta-analysis of 17 studies found that leprosy was associated with male gender, performing manual labour, ever food-shortage, sharing household with another leprosy patient, and communal living with a minimum of 5 occupants [36]. Moreover, risk factors for leprosy-related disabilities include advanced age, male, low education, multibacillary leprosy type, the presence of leprosy reactions, nerve damage, delayed diagnosis, and treatment dropout [37–42].

Findings related to demographic factors in this study, which seemed to affect the domains of ICF disability, have been observed in other studies. Most of the patients in this study were males within the range of productive age group, as also found in other studies from Indonesia, Brazil, and India [43–46]. Our results also showed that body function, activity limitation, and participation restriction had positive correlations with age. While age might not be the direct cause of disability, the decline in physical function and the escalation of physical disability associated with aging are closely intertwined with the onset of one or more diseases, including cardiovascular diseases and osteoarthritis [47]. The higher male population in leprosy cases could be attributed to differences in sociocultural behaviors between genders and life choices that would lead to a higher risk of contracting the infection [48]. More than half of this study population also had low educational attainment, indicating a shared context of socioeconomic vulnerability. This trend has been observed in Indonesian, Indian, and Brazilian studies by Menaldi et al.[8], Karotia et al. [49], and Dergan et al. [46], respectively. Individuals possessing higher levels of education are believed to adopt health-conscious lifestyles, actively acquire medical knowledge, and mitigate the risk of disability [50]. Unfortunately, diagnostic time points and treatment dropout were not explored.

Leprosy patients with multi-bacillary type demonstrate more pronounced physical dysfunction and experience greater bodily pain when compared to the pauci-bacillary group, which primarily stems from nerve impairment in leprosy patients [51]. Multi-bacillary patients were significantly at higher risk of developing nerve-function impairment [52]. Additionally, a study by Lustosa et al. (2011) revealed that multibacillary patients were associated with lower SF-36 scores across four domains: physical functioning, bodily pain, general health perceptions, and social functioning. These patients were also at higher risk of being found with grade II disability at the time of diagnosis [53].

Increased nerve thickening has been found to play a role in delaying diagnosis and potentially serves as a risk factor for deformity development [54]. Deformities were also more often seen in those with more than 3 thickened nerves. Our bivariate analyses found that the number of nerve enlargement, functional activity limitation, and social participation restriction in leprosy patients are independent of each other, which was represented by the SALSA scale ($r = 0.165$, $p = 0.044$) and P-scale ($r = 0.167$, $p = 0.041$), respectively. Following Hinkel's criteria, when the correlation coefficient falls below 0.30, it signifies a minimal or negligible association between the two variables [35]. It is worth noting that previous research has indicated that abnormalities and neuropathic pain can manifest even in the absence of nerve enlargement [55,56]. Consequently, functional activity limitations and social participation restrictions may persist even in the absence of observable nerve enlargement.

There were 29.33% of participants who experienced mild (27.33%), moderate (1.33%), and extreme (0.67%) activity limitations. This proportion is lower than that reported by Shivanna

et al. [57] in India, Nascimento et al. [58] in Brazil, and Abdela et al. [59] in Ethiopia. In two studies (Shivanna et al. [57] and Nascimento et al. [58]), most participants only felt mild functional limitations. A high proportion of patients with activity limitations may indicate prior delayed diagnosis, inadequate management, and lack of physical rehabilitation service [57–59].

Patients in this study who had limitations in social participation accounted for only one-eighth of the total participants, with 6.67% experiencing mild restriction, 4% experiencing moderate restriction, and 0.67% experiencing extreme restriction. These figures are notably lower than those found in studies conducted in Brazil, Ethiopia, and Nigeria, which reported participation restriction rates of 24%, 55.1%, and 89.3%, respectively [58–60]. Variations in cultural characteristics, socioeconomic conditions, stigma, and availability of rehabilitative services may account for these differences [60].

Functional limitations and societal restrictions are impacted by many factors. According to a previous study by Nogueira in Fortaleza, most elderly people with leprosy can still do basic daily tasks such as opening bottles with screw caps, cooking, carrying heavy objects, and walking on uneven ground [61]. Stigma and prejudice associated with leprosy can lead to societal exclusion even in the absence of visible lesions or impairments [58]. Several factors such as self-stigmatization, activity limitations, family-related issues, poverty, low education levels, inadequate rehabilitation services that include community-based rehabilitation programs, and community ignorance about the disease and its transmission can affect societal participation restriction [59].

This study also found significant differences in SALSA scores according to the degree of WHO disability for hands and feet ($p = 0.0012$ and $p = 0.0003$, respectively). Similar findings from de Souza et al. (2016) indicated that SALSA scores are associated with the degree of impairment ($p < 0.01$) [30].

There were significant differences between the total participation scale and hand disability based on WHO disability grading ($p = 0.0087$). Grade 2 disabilities, particularly those affecting the hands and especially with deformities and resulting amputations can cause discomfort and limit patients' ability to interact with their environments, participate in community activities, and pursue employment opportunities. This may lead to reduced autonomy outside the home and interference with family roles [62].

Bivariate analyses also found significant positive correlations between JTT in both dominant and non-dominant hands, and SALSA scores, including the work and dexterity domains, suggesting that reduced hand function is paralleled with limited functional activity. Meanwhile, the TUG test was used to determine functional mobility, which is typically reduced in people with foot disability [63]. There were positive correlations between TUG test and both SALSA and mobility domain of SALSA scale. Hands and feet functions are largely influenced by muscle strength, and in a prior examination utilizing data derived from the baseline of the present investigation, it was demonstrated that diminished muscular strength exhibited an association with increased self-reported challenges in the execution of routine physical activities. The encountered difficulties in the performance of daily activities exhibited a correlation with a reduction in the frequency of engagement in said activities. Furthermore, a diminished level of physical activity was identified as a predictor of subsequent decline in muscle strength [64]. Consequently, individuals with impaired muscular strength may exhibit heightened susceptibility to injurious accidents, potentially compromising their recuperative capacity following acute illnesses, injuries, or surgical interventions [64]. TUG test and total participation scale denoted no significant correlation with $p = 0.3956$. There were also no significant differences in the total participation scale between different WHO foot disability grades ($p = 0.0515$).

We found an interesting trend where hand function affected participation restriction whereas foot did not. In an older study of 63 patients with hereditary motor and sensory

neuropathy (HMSN) or Charcot-Marie-Tooth disease (CMT), upper limb disability, and not lower extremity, was considered to correlate independently with participation restriction [65]. The latter has been linked to mortality and morbidity with age. We observe a similar trend of association between hand function and societal participation in a recent study by Akbarfahimi et al. (2021) among 84 stroke patients [66]. Upper limb function measured using the Fugl-Meyer scale was found to correlate significantly with participation score (r = 0.315, p = 0.003). In an older study with 2291 healthy participants above 50 years of age, lower extremity strength and balance were not significantly associated with societal participation [67]. However, the odds of having self-reported limitation in social participation were higher in those with slower gait speed or less than 1 m/s (odds ratio [OR] = 3.1 99% confidence interval [CI] 1.5–6.2) compared to those above. Some studies have even shown supporting evidence of a relationship between gait speed and longer life. However, none of the studies discussed the reason behind this trend. Future studies detailing analysis of upper and lower extremity functions and disabilities are needed.

Most of the population in Likupang work as fishermen, which may be the cause of higher participation rates compared to studies in other countries [68,69]. Since fishermen's work is rather self-reliant, they are not affected by the stigma around them. However, fishermen required fully functioning hands to maintain their jobs. Most of them use handline fishing, using hand grip and strength to be used. In addition to fishing, the people of Likupang also work in other industries as well, such as farming and construction workers [70,71]. The main occupation in Lembata Island is the corn farming industry and most of the farmers spend around 60% of their working time on the field [70,72]. Working in this industry demands people to have high-functioning hands, due to its labor characterization of this job [73]. Farmers and fishermen with leprosy will suffer greatly mainly because of their incapability to work and use their tools properly. This study showed a significant correlation between hand disability and participation scale, which includes occupational activities such as fishing and farming.

Based on the ICF model, environmental and personal factors are also important components of disability. Risk factors for leprosy in Indonesia include floor height, house ventilation area, house window usage habits, floor and wall types, environmental sanitation, history of household contact, residential density, humidity, and economic status [74–76]. Humidity poses a significantly increased risk, with an 8.415-fold greater likelihood, of leprosy occurrence within the community. Similarly, personal hygiene presents a 6.926-fold higher risk of leprosy occurrence in the community. Residential density in Indonesian society is associated with a substantially elevated risk, estimated to be 5.754 times greater, of experiencing leprosy [76].

The populations in Lewoleba and Likupang were unequally distributed with several areas exhibiting higher density than others. There was also inadequate access to clean water, insufficient drainage, and sanitation systems, and relatively high levels of humidity, all of which are risk factors for leprosy as well as posing further challenges in managing leprosy [77,78]. Environmental conditions, personal hygiene, as well as with water resources, and management are all important factors in infectious disease transmission because they can alter the host's susceptibility along with the vector, reservoir, and exposure route of the disease [79]. One possible approach to preventing leprosy in remote and rural areas is by providing public education on maintaining the physical condition of houses under established standards. This includes ensuring the presence of ceilings, utilizing easily cleanable flooring materials, maintaining comfortable humidity levels, and limiting the occupancy of bedrooms to a maximum of two individuals [80]. Conversely, marital status and proximity to healthcare facilities do not significantly affect the likelihood of leprosy-related disability [39].

## Limitations

This study has several limitations. Firstly, correlation analysis has its pitfalls in defining a relationship between two variables, such as: the results cannot be interpreted as causal or agreement between methods, the method only incorporate two variables, and the result excludes the influence of confounding variables [81]. Therefore, we have applied confounder adjustment methods such as binary logistic regression and multivariable regression analysis to provide more objective results. We acknowledge that the multivariable linear regression models could not fit for zero collinearity assumption due to analysing variables within a variable (SALSA-work, -dexterity, and -mobility within total SALSA scale) and thus may become overfitting. However, inclusion of these variables together is considered essential in reflecting activity limitation in the study population. Another drawback from our model is that the linear assumption was not met. However, transformation of the result, which is also beyond the scope of this study, requires removal of outliers and thus may create selection bias. Secondly, this study only included the patients who were willing to participate and excluded those who would potentially be more reclusive and isolated. Although introducing more sample subjects can potentially change the results, there is a doubt that it will be a dramatic shift of trend since we have also seen similar results from other studies [30,39,41,66,67]. Thirdly, the population included in the study also came from two different areas, each with unique social and environmental characteristics. There may be some gaps in demographical factors, which we cannot currently analyse, that would influence the outcome of the study. Finally, although we found that the personal factors identified in our study such as age, gender, and educational background significantly contribute to leprosy-related disability, other influential factors including personal hygiene, contact history, socio-economic status, health beliefs, and religion were not specifically examined in our research [75,82–84], highlighting the need for further exploration and investigation.

## Conclusion

Through this study, we utilized ICF to identify factors relating to leprosy patients' disability progression in Indonesia. Although we found mild to independent correlations between the aspects of ICF in this study population, comparison analyses showed that several personal factors such as age and education could impact disability development in leprosy patients through their actions towards activity limitation and body functions. In addition, participation restrictions also affected the development of disability. We recommend clinicians to address all these factors to adjust their treatment approach as a preventive measure for the development of disability in leprosy patients and use ICF as part of routine assessment standard. It will also be beneficial to make ICF as part of National Registered System. This study also emphasizes the importance of equal access to primary and secondary education in Indonesia, especially in remote areas. We suggest conducting future exploration studies using factorial analysis and nonlinear regression models of the ICF factors in a greater number of leprosy subjects residing in more diverse areas of the country. We also suggest conducting more research to study the impact of educational intervention, by developing methods to adjust to the population's educational levels by using interactive videos and pictures in rural populations, towards disease and disability prevention.

## Supporting information

**S1 Data. Raw data of the study population for Table 2 (demographics of leprosy patients) and Table 3 (bivariate analysis of age and other variables).**
(XLSX)

**S2 Data. Raw material for comparison of Total SALSA scores between different educational level in the study population (Fig 3A).**
(XLSX)

**S3 Data. Raw material for comparison of SALSA-Work scores between different educational level in the study population (Fig 3B).**
(XLSX)

**S4 Data. Raw material for comparison of JTT test in dominant hand in seconds between different educational level in the study population (Fig 3C).**
(XLSX)

**S5 Data. Raw material for comparison of JTT test in non-dominant hand in seconds between different educational level in the study population (Fig 3D).**
(XLSX)

**S6 Data. Raw material for comparison of TUG in seconds between different educational level in the study population (Fig 3E).**
(XLSX)

**S7 Data. Raw material for comparison Total SALSA scores in the study population according to WHO grading of hand disability (Fig 4A).**
(XLSX)

**S8 Data. Raw material for comparison SALSA-Work scores in the study population according to WHO grading of hand disability (Fig 4B).**
(XLSX)

**S9 Data. Raw material for comparison SALSA-Dexterity scores in the study population according to WHO grading of hand disability (Fig 4C).**
(XLSX)

**S10 Data. Raw material for comparison of P-scale in the study population according to WHO grading of hand disability (Fig 4D).**
(XLSX)

**S11 Data. Raw material for comparison of JTT in the dominant hand in the study population according to WHO grading of hand disability (Fig 4E).**
(XLSX)

**S12 Data. Raw material for comparison of JTT in the non-dominant hand in the study population according to WHO grading of hand disability (Fig 4F).**
(XLSX)

**S13 Data. Raw material for comparison of Total SALSA scores in the study population according to WHO grading of foot disability (Fig 5A).**
(XLSX)

**S14 Data. Raw material for comparison of SALSA-Mobility scores in the study population according to WHO grading of foot disability (Fig 5B).**
(XLSX)

**S15 Data. Raw material for comparison of P-scale in the study population according to WHO grading of foot disability (Fig 5C).**
(XLSX)

**S16 Data. Raw material for comparison of Total SALSA, SALSA-mobility, -work, -dexterity scores, P-scale, JTT in both hands in seconds, and TUG in seconds in the study population (Fig 6).**
(XLSX)

**S17 Data. Multivariable regression analyses of significantly correlated ICF variables (body function, activity, and participation).**
(XLSX)

## Acknowledgments

This work is supported by the Directorate of Research and Community Engagement of Universitas Indonesia. We would like to thank the directors and staffs of St. Damian Hospital Lewoleba-Lembata-East Nusa Tenggara, Ministry of Health East Lewoleba-East Nusa Tenggara, Ministry of Health North Minahasa-North Sulawesi, and Faculty of Medicine, University of Sam Ratulangi, Manado City-North Sulawesi. We would like to acknowledge the KATA-MATAKU team and the medical students of Universitas Indonesia.

## Author Contributions

**Conceptualization:** Luh Karunia Wahyuni, Nelfidayani Nelfidayani, Melinda Harini, Fitri Anestherita, Rizky Kusuma Wardhani.

**Data curation:** Luh Karunia Wahyuni, Sri Linuwih Menaldi, Yunia Irawati, Gitalisa Andayani, Hisar Daniel.

**Formal analysis:** Nelfidayani Nelfidayani, Melinda Harini, Intan Savitri, Petrus Kanisius Yogi Hariyanto, Isabela Andhika Paramita.

**Funding acquisition:** Luh Karunia Wahyuni, Sri Linuwih Menaldi, Yunia Irawati.

**Investigation:** Luh Karunia Wahyuni, Nelfidayani Nelfidayani, Melinda Harini, Fitri Anestherita, Rizky Kusuma Wardhani, Sri Linuwih Menaldi, Yunia Irawati, Tri Rahayu, Gitalisa Andayani, Hisar Daniel.

**Methodology:** Luh Karunia Wahyuni, Nelfidayani Nelfidayani, Melinda Harini, Fitri Anestherita, Rizky Kusuma Wardhani.

**Project administration:** Luh Karunia Wahyuni, Nelfidayani Nelfidayani, Melinda Harini, Fitri Anestherita, Rizky Kusuma Wardhani, Sri Linuwih Menaldi, Yunia Irawati, Tri Rahayu, Intan Savitri, Petrus Kanisius Yogi Hariyanto, Isabela Andhika Paramita.

**Resources:** Luh Karunia Wahyuni, Melinda Harini, Fitri Anestherita, Sri Linuwih Menaldi, Yunia Irawati, Gitalisa Andayani, Hisar Daniel.

**Software:** Nelfidayani Nelfidayani, Rizky Kusuma Wardhani, Tri Rahayu, Intan Savitri, Petrus Kanisius Yogi Hariyanto, Isabela Andhika Paramita.

**Supervision:** Luh Karunia Wahyuni, Fitri Anestherita, Sri Linuwih Menaldi, Yunia Irawati, Intan Savitri, Petrus Kanisius Yogi Hariyanto, Isabela Andhika Paramita.

**Validation:** Nelfidayani Nelfidayani, Melinda Harini, Fitri Anestherita, Rizky Kusuma Wardhani.

**Visualization:** Sri Linuwih Menaldi, Yunia Irawati, Tri Rahayu, Gitalisa Andayani, Hisar Daniel, Intan Savitri, Petrus Kanisius Yogi Hariyanto, Isabela Andhika Paramita.

**Writing – original draft:** Luh Karunia Wahyuni, Nelfidayani Nelfidayani, Melinda Harini, Fitri Anestherita, Rizky Kusuma Wardhani.

**Writing – review & editing:** Luh Karunia Wahyuni, Nelfidayani Nelfidayani, Melinda Harini, Fitri Anestherita, Rizky Kusuma Wardhani, Sri Linuwih Menaldi, Yunia Irawati, Tri Rahayu, Gitalisa Andayani, Hisar Daniel, Intan Savitri, Petrus Kanisius Yogi Hariyanto, Isabela Andhika Paramita.

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
