## [Decision Letter · Decision Letter 0]

19 Aug 2023

Dear Wahyuni,

Thank you very much for submitting your manuscript "The International Classification of Functioning, Disability and Health to map leprosy-related disability in rural and remote areas in Indonesia" for consideration at PLOS Neglected Tropical Diseases. As with all papers reviewed by the journal, your manuscript was reviewed by members of the editorial board and by several independent reviewers. In light of the reviews (below this email), we would like to invite the resubmission of a significantly-revised version that takes into account the reviewers' comments. 

We cannot make any decision about publication until we have seen the revised manuscript and your response to the reviewers' comments. Your revised manuscript is also likely to be sent to reviewers for further evaluation.

Sincerely,

Alberto Novaes Ramos Jr

Academic Editor

Stuart Blacksell

Section Editor

Reviewer's Responses to Questions

**Key Review Criteria Required for Acceptance?**

**Methods**

-Are the objectives of the study clearly articulated with a clear testable hypothesis stated?

-Is the study design appropriate to address the stated objectives?

-Is the population clearly described and appropriate for the hypothesis being tested?

-Is the sample size sufficient to ensure adequate power to address the hypothesis being tested?

-Were correct statistical analysis used to support conclusions?

-Are there concerns about ethical or regulatory requirements being met?

Reviewer #1: (No Response)

Reviewer #2: The study objective is clearly described in the end of the introduction section.

The study design is appropriate to clarify the research question. 

The inclusion criteria is well stated and was suitable for the study goals. 

Sample size is sufficient and the statistical analysis is appropriate.

The study has been approve by the Institutional Review Board (IRB) and patients gave their consent prior to the study.

Reviewer #3: the objectives of the study needs to more clarified , adjust with the result part, whilst the study design, population are clearly describe, so does for sufficient sample size (but state the minimal sample size needed). There are not concerns about ethical or regulatory issues for this article

Reviewer #4: -The objectives of the study clearly articulated 

-The study design appropriate to address the stated objectives.

-The population clearly described but what criteria for diagnosis methods used to select the study population( leprosy cases) were not mentioned properly .

-The method of sample size calculation and sampling technic were not properly explained . Why they choice two study areas ? why not more than two areas to increase the sample size ( the reasons were not explained properly. 

-Fisher's exact test and chi-square test are statistical methods commonly used to analyze categorical variables .These tests can determine whether there is a statistically significant relationship between two categorical variables. However, it's important to note that these tests themselves do not control for confounding factors, these leading to a spurious association. To control for confounding factors in the analysis, additional methods are required for effectively controlling confounding factors and drawing valid conclusions.

-Ethical or regulatory requirements being met properly

**Results**

-Does the analysis presented match the analysis plan?

-Are the results clearly and completely presented?

-Are the figures (Tables, Images) of sufficient quality for clarity?

Reviewer #1: (No Response)

Reviewer #2: The results were presented according to the analysis plan. 

The figures and tables were clear and sufficiently presented.

Reviewer #3: The result and the figures are presented clearly, sufficient enouth, but some of analysis result do not quite match with the analysis plan, should clarify and detail in aims/objectives of the study.

Reviewer #4: -The analysis presented match with the analysis plan. 

-why you used mean age ? were the age data normal distributed? 

- The figures and the tables are used properly but not self explanatory (where, what , when ...)

**Conclusions**

-Are the conclusions supported by the data presented?

-Are the limitations of analysis clearly described?

-Do the authors discuss how these data can be helpful to advance our understanding of the topic under study?

-Is public health relevance addressed?

Reviewer #1: (No Response)

Reviewer #2: The conclusions were supported by the data.

Study limitations have been described clearly.

Reviewer #3: the conclusion matched with the result, so do the limitations, but maybe need more detail on objectives of the study . Authors are suggested to put more on discussion so that the readers will understand better , but publich health relevance is strongly addresed

Reviewer #4: The conclusions supported by the data presented and the limitations clearly described

-Do the authors discuss how these data can be helpful to advance our understanding of the topic under study? Yes

-Is public health relevance addressed? yes

**Editorial and Data Presentation Modifications?**

Reviewer #1: Introduction:

 In the introduction, reconcile the content of the research problem (lines 98 to 100) with the objectives (lines 132 to 135) so that the research objective is clear and uniform. Check if there is indeed a need for maps to contextualize the study, and if necessary, review the possibility of presenting them in the methodology. Present the testable hypothesis in the study more clearly. 

Materials and methods: 

In materials and methods, present the procedures adopted for the protection and secrecy of the identity of the participants. In the "study design and population", include the total number of patients approached, the exclusion criteria (example: neuropathies not associated with leprosy, previous disabilities, mental disabilities, children, among others). In the inclusion criteria, make it clear “who the leprosy patients are, that is, time of diagnosis, if the patients are undergoing treatment or if they are undergoing post-treatment rehabilitation; these factors may interfere with the results found. In the study protocol, justify the reason for grouping the data from the two regions together in the results, since they are different areas, with different characteristics. Describe better how the questionnaire was applied, whether it was by a previously trained professional, or the patient who filled it out, for example. In Figure 2, the variables shown in the table, "occupation, personal hygiene and socioeconomic status", were addressed in the discussion, but were not shown in the results (if this data has not been researched, remove it from the model). Correct the title of the Figure. 4 (it was numbered as Figure 5). 

Conclusion:

In conclusion, review whether the following information is consistent with the results: Activity limitation (...) is lower in patients with more severe hand impairment.

Reviewer #2: Vertical lines in the table should be deleted.

Reviewer #3: minor modification , adjust the objectives of this study

Reviewer #4: The paper did good way but it required Minor revisions

**Summary and General Comments**

Reviewer #1: The article is of great importance, being of extreme adoration in the health area. It uses internationally validated and recognized disability classification tools, as well as the intuitive one among the instruments, further strengthening the methodology used. The abstract contextualizes well what will be understood in the research, and the introduction allows to support the research problem. Tables and figures were used in the methodology that allowed a better understanding of the tools and the design of the study, being a strong point of the research. In the methodology, as a point of improvement, it is necessary to strengthen the description of the population. The discussion serves the purpose of strengthening the results found. I suggest that a future study includes the participation of nurses, relevant professionals in research and care for leprosy patients.

Reviewer #2: Overall, the data was helpful to improve our understanding regarding the impact of leprosy-related disability in the patients body function, activity, and participation.

Reviewer #3: The strength of this study is that this study reveals some findings of community in spesific area that have high prevalence, but the aims/ objectives of this study need more to be clarified and adjusted with the result.

Reviewer #4: The paper is well done with some limitations , what criteria for diagnosis methods used to select the study population( leprosy cases) were not mentioned properly , sample size calculation and sampling technic were not properly explained ,To control for confounding factors in the analysis, additional methods are required for effectively controlling confounding factors and drawing valid conclusions, figures and the tables are not self explanatory , the the references numbers in the paper were putted after the full stop. The paper could not follow the steps of the Journal guideline.

PLOS authors have the option to publish the peer review history of their article (what does this mean?). If published, this will include your full peer review and any attached files.

Reviewer #1: No

Reviewer #2: No

Reviewer #3: No

Reviewer #4: No
---

## [Decision Letter · Decision Letter 1]

29 Nov 2023

Dear Wahyuni,

Thank you very much for submitting your manuscript "The International Classification of Functioning, Disability and Health to Map Leprosy-Related Disability in Rural and Remote Areas in Indonesia" for consideration at PLOS Neglected Tropical Diseases. As with all papers reviewed by the journal, your manuscript was reviewed by members of the editorial board and by several independent reviewers. The reviewers appreciated the attention to an important topic. Based on the reviews, we are likely to accept this manuscript for publication, providing that you modify the manuscript according to the review recommendations. 

Sincerely,

Alberto Novaes Ramos Jr

Academic Editor

Stuart Blacksell

Section Editor

Reviewer's Responses to Questions

**Key Review Criteria Required for Acceptance?**

**Methods**

-Are the objectives of the study clearly articulated with a clear testable hypothesis stated?

-Is the study design appropriate to address the stated objectives?

-Is the population clearly described and appropriate for the hypothesis being tested?

-Is the sample size sufficient to ensure adequate power to address the hypothesis being tested?

-Were correct statistical analysis used to support conclusions?

-Are there concerns about ethical or regulatory requirements being met?

Reviewer #1: The methodology is clearly described. I have no further considerations on this topic.

Reviewer #2: (No Response)

Reviewer #3: Yes , the objectives are clearly stated, appropriate study design, the sample size is sufficient and the authors used correct statistical analysis. There is no concern about ethical

Reviewer #4: -Are the objectives of the study clearly articulated with a clear testable hypothesis stated? yes

-Is the study design appropriate to address the stated objectives?yes

-Is the population clearly described and appropriate for the hypothesis being tested? yes

-Is the sample size sufficient to ensure adequate power to address the hypothesis being tested? yes

-Were correct statistical analysis used to support conclusions? yes

-Are there concerns about ethical or regulatory requirements being met? yes

**Results**

-Does the analysis presented match the analysis plan?

-Are the results clearly and completely presented?

-Are the figures (Tables, Images) of sufficient quality for clarity?

Reviewer #1: The results are clearly described. I have no further considerations on this topic.

Reviewer #2: (No Response)

Reviewer #3: Yes, the analysis present 

some results shown in the figure are suggested to make it clearer, such as which one is statistically significant, you can put the symbol in the figure, not only by words and state what the symbol * means (Figure 3,4 and 5)

for figure 6, I suggest please to put statement which variables show the strongest correlation (r) and also the range of coefficient correllation

Reviewer #4: The analysis and presentation of results match the analysis plan. Results are well-presented with quality of figures and tables

**Conclusions**

-Are the conclusions supported by the data presented?

-Are the limitations of analysis clearly described?

-Do the authors discuss how these data can be helpful to advance our understanding of the topic under study?

-Is public health relevance addressed?

Reviewer #1: The conclusion is clear and described based on the results.

Reviewer #2: (No Response)

Reviewer #3: Yes, the conclusion supported with the data, limitation of the study are stated, eventhough not for limitation of analysis. Public Health relevace is addressed, but as suggestion, it would be better if the author mention what kind of educational and disablitiy prevention

Reviewer #4: The conclusions drawn by the authors are well-supported by the data. However, it would be valuable to include deeper clinical implications and recommendations for healthcare professionals and policy developers

**Editorial and Data Presentation Modifications?**

Reviewer #1: Thank you for considering the suggested changes. The change in graphs/tables made the results clearer. I have no further considerations on this topic.

Reviewer #2: (No Response)

Reviewer #3: minor revision

Reviewer #4: Minor revision

**Summary and General Comments**

Reviewer #1: The authors made the suggested changes, which improved this version. I have no additional suggestions for changes to the article.

Reviewer #2: (No Response)

Reviewer #3: The article is really interesting, it shows the analysis about functional capacity of the leprosy patients and detailed analysis is shown as a strength of this article. Though it has several limitations stated by the author, but since the data from the location with high prevalence of leprosy, it shows the uniqueness of this study. The community engagement program (KATAMATAKU) is great.

Reviewer #4: The study design was well-executed and the methods were clearly described. However, the sample size could have been larger to increase generalizability

The statistical analysis appears to be sound. However, it would be helpful to include confidence intervals for the reported results

It would be beneficial to provide more detailed information about participant demographics

The authors included more recent literature to strengthen the background and relevance of the study.

The discussion section provides a comprehensive analysis and interpretation of the findings. However, it would be helpful to discuss potential implications for future research or clinical practice.

The paper is well-written and organized. However, some sections could be better clarified, especially in terms of terminology

The study results are significant and contribute to the existing literature. However, it would be valuable to include a discussion on the potential mechanisms underlying the observed associations

The conclusions drawn by the authors are well-supported by the data. However, it would be valuable to include deeper clinical implications and recommendations for healthcare professionals and policy developers

PLOS authors have the option to publish the peer review history of their article (what does this mean?). If published, this will include your full peer review and any attached files.

Reviewer #1: No

Reviewer #2: No

Reviewer #3: No

Reviewer #4: No

Figure Files:

Data Requirements:

Reproducibility:

References

---

## [Decision Letter · Decision Letter 2]

3 Feb 2024

Dear Wahyuni,

Thank you very much for submitting your manuscript "The International Classification of Functioning, Disability and Health to Map Leprosy-Related Disability in Rural and Remote Areas in Indonesia" for consideration at PLOS Neglected Tropical Diseases. As with all papers reviewed by the journal, your manuscript was reviewed by members of the editorial board and by several independent reviewers. The reviewers appreciated the attention to an important topic. Based on the reviews, we are likely to accept this manuscript for publication, providing that you modify the manuscript according to the review recommendations. 

Sincerely,

Alberto Novaes Ramos Jr

Academic Editor

Stuart Blacksell

Section Editor

Reviewer's Responses to Questions

**Key Review Criteria Required for Acceptance?**

**Methods**

-Are the objectives of the study clearly articulated with a clear testable hypothesis stated?

-Is the study design appropriate to address the stated objectives?

-Is the population clearly described and appropriate for the hypothesis being tested?

-Is the sample size sufficient to ensure adequate power to address the hypothesis being tested?

-Were correct statistical analysis used to support conclusions?

-Are there concerns about ethical or regulatory requirements being met?

Reviewer #1: The methodology is clearly described. I have no further considerations on this topic

Reviewer #2: (No Response)

Reviewer #3: Yes, the objectives of the study are clearly stated, appropriate study design and population, enough sample size and proper statistical analysis 

There are no ethical issue on this article

Reviewer #4: • The study design clearly

• The inclusion and exclusion criteria well-defined

• The study population clearly described, including demographics and any relevant baseline characteristics

• The exposure and outcome variables clearly defined

• The data collection methods, including any tools or instruments used, well-explained

• A detailed description of the statistical methods used for data analysis but not give explanation for neglecting the potential influence of other variables.

**Results**

-Does the analysis presented match the analysis plan?

-Are the results clearly and completely presented?

-Are the figures (Tables, Images) of sufficient quality for clarity?

Reviewer #1: The results are clearly described. I have no further considerations on this topic.

Reviewer #2: (No Response)

Reviewer #3: Yes, revisions from the author have fulfilled the criteria

Reviewer #4: • the results presented clearly and concisely but the prevalence better put in 95% CI (, ),especially for 82% of subjects with multibacillary leprosy, 10.67% of subjects with grade 2 WHO hand disability, and 9.33% of subjects with grade 2 WHO foot disability.

• there appropriate use of tables and figures to display data but some table (table 2 result showed mathematical error for variable WHO Hand Disability 150 but 148

• statistical analyses appropriate, and are the results interpreted correctly with missing confounding factors

• potential confounding factors not addressed adequately

Bivariate analyses provide a limited perspective on the complexity of real-world phenomena. To gain a more comprehensive understanding, researchers often need to employ multivariate analyses that consider multiple variables simultaneously. Bivariate analyses have certain limitations and drawbacks that should be considered when interpreting their results. Here are some common drawbacks:

Bivariate analyses focus on the relationship between two variables only, neglecting the potential influence of other variables. Real-world phenomena are often influenced by multiple factors, and the exclusion of relevant variables can oversimplify the understanding of the relationship.

Bivariate analyses may fail to account for confounding variables, which are third variables that can affect the observed relationship between the two variables being studied. Without controlling for these confounding variables, the true relationship between the variables of interest may be distorted.

Bivariate analyses, particularly correlation analyses, can show the strength and direction of a relationship but do not establish causation. Correlation does not imply causation, and there may be underlying factors or unobserved variables influencing both variables in the analysis.

**Conclusions**

-Are the conclusions supported by the data presented?

-Are the limitations of analysis clearly described?

-Do the authors discuss how these data can be helpful to advance our understanding of the topic under study?

-Is public health relevance addressed?

Reviewer #1: The conclusion is clear and described based on the results.

Reviewer #2: (No Response)

Reviewer #3: Yes, the authors have already described the limitation of the study , so is the public health relevance

Reviewer #4: well addressed but The authors should clearly discuss the limitations associated with the smaller sample size and how these limitations might introduce the roles of chance into the results. Discussing the potential impact of these limitations on the generalizability and interpretation of the findings would be valuable with explanation of analysis

**Editorial and Data Presentation Modifications?**

Reviewer #1: I have no further considerations on this topic.

Reviewer #2: (No Response)

Reviewer #3: Accept

Reviewer #4: Minor Revision

**Summary and General Comments**

Reviewer #1: I have no additional suggestions for changes to the article.

Reviewer #2: (No Response)

Reviewer #3: this paper is very beneficial, especially the research was done in the place where the prevalence of the disease is high. Through this paper, we could get description and future recommendation to solve this health problem

Reviewer #4: well done. but the analysis and limitation require some modification

PLOS authors have the option to publish the peer review history of their article (what does this mean?). If published, this will include your full peer review and any attached files.

Reviewer #1: No

Reviewer #2: No

Reviewer #3: No

Reviewer #4: No

Figure Files:

Data Requirements:

Reproducibility:

References

---

## [Editor Report · Decision Letter 3]

29 Mar 2024

Dear Wahyuni,

We are pleased to inform you that your manuscript 'The International Classification of Functioning, Disability and Health to Map Leprosy-Related Disability in Rural and Remote Areas in Indonesia' has been provisionally accepted for publication in PLOS Neglected Tropical Diseases.

Best regards,

Stuart D. Blacksell

Section Editor

---

## [Editor Report · Acceptance letter]

8 May 2024

Dear Wahyuni,

We are delighted to inform you that your manuscript, "The International Classification of Functioning, Disability and Health to Map Leprosy-Related Disability in Rural and Remote Areas in Indonesia," has been formally accepted for publication in PLOS Neglected Tropical Diseases.

Best regards,

Shaden Kamhawi

co-Editor-in-Chief

Paul Brindley

co-Editor-in-Chief
